# What Limits Bidirectional Model's Generative Capabilities?
# A Uni-Bi-Directional Mixture-of-Expert Method For Bidirectional Fine-tuning

**Zuchao Li** [1]  **Yonghua Hei** [2]  **Qiwei Li** [3]  **Lefei Zhang** [3]  **Ping Wang** [4]  **Hai Zhao** [5]  **Baoyuan Qi** [6]  **Guoming Liu** [6]

## Abstract

Large Language Models (LLMs) excel in generation tasks, yet their causal attention mechanisms limit performance in embedding tasks. While bidirectional modeling may enhance embeddings, naively fine-tuning unidirectional models bidirectionally severely degrades generative performance. To investigate this trade-off, we analyze attention weights as dependence indicators and find that bidirectional fine-tuning increases subsequent dependence, impairing unidirectional generation. Through systematic Transformer module evaluations, we discover the FFN layer is least affected by such dependence. Leveraging this discovery, we propose UBMoE-LLM, a novel Uni-Bi-directional Mixture-of-Experts LLM, which integrates the original unidirectional FFN with a bidirectionally fine-tuned FFN via unsupervised contrastive learning. This MoE-based approach enhances embedding performance while preserving robust generation. Extensive experiments across diverse datasets and model scales validate our attention dependence metric and demonstrate UBMoE-LLM's superior generative quality and reduced hallucination. Code is available at: https://github.com/heiyonghua/ubmoe_llm.

## 1. Introduction

Creating a model capable of both generation and text embedding has been a long-term goal in the field of natural

---

[1]School of Artificial intelligence, Wuhan University [2]This work was conducted during Yonghua Hei's internship at the School of Artificial Intelligence, Wuhan University. [3]School of Computer Science, Wuhan University [4]School of Information Management, Wuhan University [5]Department of Computer Science and Engineering, Shanghai Jiao Tong University [6]Xiaomi, Beijing, China. Correspondence to: Lefei Zhang <zhanglefei@whu.edu.cn>, Ping Wang <wangping@whu.edu>.

*Proceedings of the 42nd International Conference on Machine Learning*, Vancouver, Canada. PMLR 267, 2025. Copyright 2025 by the author(s).

language processing (Muennighoff et al., 2024; Bao et al., 2020; Dong et al., 2019). Large causal language models have shown strong capabilities across multiple tasks (Touvron et al., 2023a;b; Jiang et al., 2023; Bai et al., 2023), making them a promising choice for achieving this goal. Recent research indicates that causal language models are limited by their causal attention mechanisms in embedding tasks, making unidirectional models suboptimal for these tasks (BehnamGhader et al., 2024; Springer et al., 2024; Muennighoff et al., 2024). A natural idea to address this issue is to enhance the bidirectional modeling capabilities of causal language models. However, simply enabling bidirectional modeling in Large Language Model (LLMs) results in a significant decline in generative performance (Muennighoff et al., 2024). Therefore, a critical challenge in building such a unified model is how to enhance its bidirectional modeling capabilities while maintaining robust generative performance.

Previous studies have explored two main methods to enhance the bidirectional modeling capabilities of unidirectional language models:

**(1) Transforming a unidirectional decoder-only language model into a bidirectional encoder model** This method initializes the weights using a unidirectional language model and enables bidirectional attention. As shown in the Bidirection in Figure 1, the tokens in the model self-attention matrix are visible to each other. Through contrastive learning (Gao et al., 2022; BehnamGhader et al., 2024; Muennighoff et al., 2024), this approach maximizes the similarity between the representations of positive samples while minimizing the similarity with negative sample representations, thereby enhancing the model's ability to contextually model bidirectional information. Behamghader et al. (BehnamGhader et al., 2024) and Li et al. (Li & Li, 2024b) use this method to turn a unidirectional decoder-only language model into a strong bidirectional encoder model. This method can make the model have a better text understanding ability, but will bring disastrous effects on the generation tasks (Muennighoff et al., 2024).

**(2) Constructing generative models with local bidirectional modeling** In the instruction fine-tuning phase,

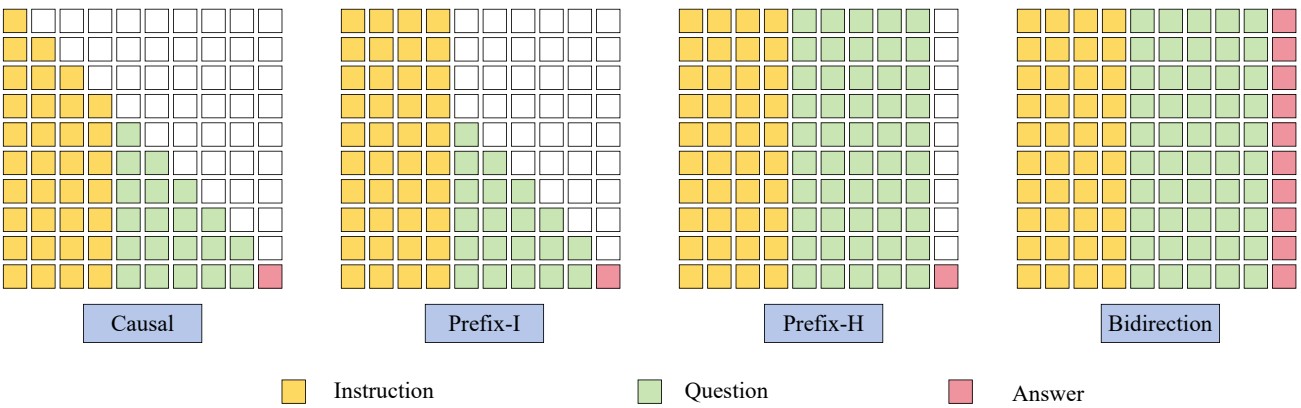

*Figure 1.* The image depicts three different attention mask mechanisms: unidirectional attention, prefix attention, and bidirectional attention. In prefix attention, bidirectional attention is used for the instruction part, while unidirectional attention is maintained for the other parts.

a local bi-directional attention mechanism is introduced, as shown in Figure 1. There are two main implementations, Prefix-I (Rafailov et al., 2023) and Prefix-H (Devlin et al., 2019; He et al., 2023). Prefix-I uses bi-directional attention in the instruction part and keeps it unidirectional in the rest of the part, while Preifx-H enables bi-directional attention for all the history conversations, and keeps the unidirectionality and calculates the loss only for the last reply. However, this approach still has some drawbacks (Muennighoff et al., 2024). Prefix-I and Prefix-H maintain unidirectionality in the response part and only compute loss in the response part. Since in the training phase, some of the tokens can see some of the later tokens, which makes it possible to increase the attention to the later tokens in the optimization process to optimize the representation of the hidden state, and this process may generate subsequent dependence. In the generation phase, the subsequent dependence it generates affects the same tokens, which makes these tokens that should be modeled unidirectionally generate an inappropriate dependence distribution.

Recent work has shown that both methods result in varying degrees of degradation in generation performance (Muennighoff et al., 2024). However, the reasons for this degradation remain unexplained. An intuitive explanation is that this degradation is caused by subsequent dependence during the training phase (Wang et al., 2022). To test this interpretation, we propose an attention dependence based explanation method which considers the attention weight of each token on other tokens as a dependence and constructs average preceding/subsequent dependence measures at the hierarchical and global levels. Previous research has shown that all generative tasks can be reduced to generating dialogues following user instructions (Raffel et al., 2023; Du et al., 2022); therefore, we use instruction texts to evaluate such dependence. We demonstrate that this effect is brought about by

enabling bi-directional attention in the training phase, from comparing the effects of different model scales, evaluation datasets, token lengths, and training methods on subsequent dependence. Through extensive experiments, we verify the correlation between subsequent attention dependence and the generation ability of the model. Besides, we find that the FFN layer is least affected in bidirectional fine-tuning.

Based on this finding, we propose a novel **Uni-Bi-Directional Mixture-of-Expert Large Language Model, UBMoE-LLM** which can reduce the damage of the subsequent dependence on generation performance while enhancing the model's performance in embeddings. We first initialize the weights using a model fine-tuned with instructions, then enable bidirectional attention and use supervised contrastive learning to enhance the model's word embedding capability, updating only the FFN parameters during this process. We use extensive comparative experiments to demonstrate the rationality of this approach. Next, we parallel the FFN layer of the word embedding fine-tuned model with the FFN layer of the original instruction fine-tuned model and use a gating mechanism to select the parameters activated for each token, constructing a mixture of experts model with two experts. Finally, we fine-tune the parameters of the gating layer using a small amount of data to improve the balance and effectiveness of expert selection.

To evaluate the effectiveness of our method across different scales of language models, we used four different scales of Qwen1.5-Chat (Bai et al., 2023) models ranging from 0.5B to 7B as the backbone models to validate our method's effectiveness. First, we experimentally verified the effectiveness of fine-tuning only the FFN layer, which achieved performance comparable to fine-tuning all parameters on ten text similarity (STS) tasks in MTEB (Muennighoff et al., 2022), consistently demonstrating superior performance. Next, we combined the instruction-tuned model and the

embedding-tuned model to construct UBMoE-LLM and fine-tuned the gate layer. We evaluated UBMoE-LLM using multiple generative criteria, and the results indicated that this approach enhances the model's embedding performance while maintaining robust generative performance. We also tested UBMoE-LLM on the TruthfulQA dataset and the results show that UBMoE-LLM incorporating bidirectional attention can effectively resist the hallucinatory problem.

## 2. Related Work

**Enhanced text embedding performance for LLM** Initially, bi-directional pre-trained language models are a robust choice for text embedding tasks (Devlin et al., 2019; Lan et al., 2020). Recent research has shown that large unidirectional language models can also be transformed into powerful encoders for embedding tasks (Li & Li, 2023). However, although large causal language models show strong performance on multi tasks (Touvron et al., 2023a;b; Jiang et al., 2023; Bai et al., 2023; Zeng et al., 2023; Brown et al., 2020), but their preference in text embedding has been limited due to its architectural (BehnamGhader et al., 2024). Recent research suggests that causal self-attention mechanism is the main reason, with its limitation of models to acquire information from the entire sequence. For this, its text embedding ability can be effectively enhanced by simply enabling bidirectional attention and using contrast learning for training (BehnamGhader et al., 2024; Li & Li, 2024b).

**Language model with both embedding and generation capabilities** Initially, the encoder-decoder model was a natural choice to unify the generation and embedding tasks (Vaswani et al., 2023; Lewis et al., 2019; Raffel et al., 2023). But it was replaced by decoder-only unidirectional language models and encoder-only bidirectional language models for generation and embedding tasks (Lan et al., 2020). Recent research has shown that how to unify unidirectional and bidirectional capabilities is key to framing such a unified model (Muennighoff et al., 2024). But the severe generative performance loss problem associated with bi-directional training makes this unification limited. So, a discussion of the causes of this damage is necessary for the construction of such a unified model.

**Interpreting language models using attention weights** Based on the self-attention mechanism, language models calculate attention weights through a dot product mechanism, allowing each token to observe other tokens with varying weights. Previous research has suggested that the interpretability of attention weights may vary significantly across different NLP tasks, and there are critical views on whether attention weights truly provide insights into model predictions (Vashishth et al., 2019; Chrysostomou & Ale-

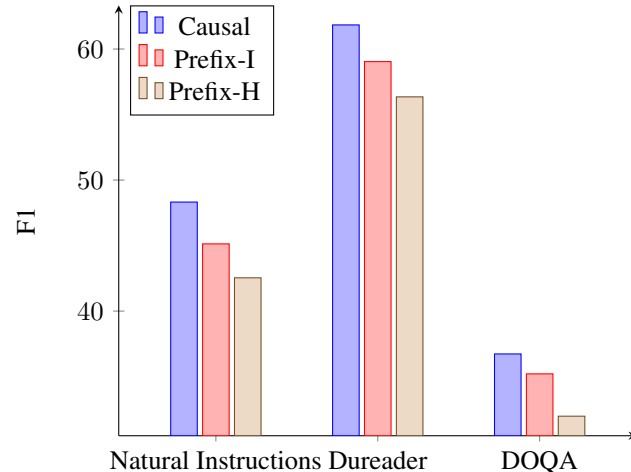

*Figure 2.* Preliminary experiments results.

tras, 2021). These earlier studies analyzed different tasks using different models due to the lack of a powerful single language model. Recent research indicates that various NLP tasks can be unified as instruction-following tasks (Du et al., 2022), and large language models have become the preferred choice for multiple tasks. This makes it possible to analyze the interpretability of attention weights from a unified model and perspective

## 3. What Causes Generative Performance Decline?

### 3.1. Preliminaries

Previous work (Ding et al., 2024) has demonstrated that overcoming the bidirectional attention-pure decoder used for text-embedding tasks has several advantages. We present the performance effects of recent work on bidirectional attention in Appendix A. Recent work (BehnamGhader et al., 2024) has shown that the supplement of bidirectional attention will lead to the degradation of model generation ability. We conducted preliminary experiments to test the impact of existing bidirectional modeling enhancement efforts.

**Generation performance.** We tested the generation task effects of *Cusual*, *Prefix-I* and *Prefix-H* methods on the Natural Instructions (Mishra et al., 2021), DOQA (Campos et al., 2020) and Dureader (Contributors, 2021) datasets. *Prefix-I* and *Prefix-H* are the bidirectional attention enhancement methods mentioned above. *Cusual* represent unidirectional attention calculations. We add a detailed experimental setup for pilot analysis. We set the batch size to 1, loss accumulation to 32, learning rate to 0.00003, and others to follow the default Settings of the transformer library (Wolf et al., 2020). We can see from the Figure 1 that from *Cuaual* to

*Table 1.* The result of fine-tuning Qwen1.5-0.5B and Qwen1.5-1.8B with the DOQA dataset. "Full test dataset" mean use all office test set to evaluate, "W/o no answer" mean no unanswerable questions in the test set.

| Model | Method | Bleu-1 | Bleu-2 | Bleu-3 | Bleu-4 | Dist-1 | Dist-2 | Dist-3 | F1 | Rouge-L |
|---|---|---|---|---|---|---|---|---|---|---|
| | | | | | Full test dataset | | | | | |
| Qwen1.5-0.5B | causal | 40.24 | 34.29 | 31.96 | 30.73 | 8.46 | 34.06 | 49.67 | 41.03 | 39.87 |
| Qwen1.5-0.5B | Prefix-I | 37.11 | 30.79 | 28.33 | 27.04 | 8.44 | 32.89 | 47.85 | 38.80 | 37.55 |
| Qwen1.5-1.8B | causal | 44.39 | 39.20 | 37.16 | 36.11 | 9.40 | 37.78 | 54.80 | 44.53 | 43.43 |
| Qwen1.5-1.8B | Prefix-I | 41.89 | 36.24 | 34.02 | 32.83 | 9.30 | 36.95 | 53.72 | 43.13 | 41.94 |
| | | | | | W/o no answer | | | | | |
| Qwen1.5-0.5B | causal | 32.62 | 26.07 | 23.82 | 22.84 | 10.14 | 39.27 | 56.00 | 28.02 | 26.36 |
| Qwen1.5-0.5B | Prefix-I | 28.95 | 22.26 | 20.01 | 19.06 | 9.73 | 36.39 | 51.43 | 25.75 | 24.05 |
| Qwen1.5-1.8B | causal | 37.33 | 31.80 | 29.85 | 29.01 | 10.99 | 42.76 | 60.86 | 33.86 | 32.38 |
| Qwen1.5-1.8B | Prefix-I | 36.85 | 31.05 | 29.02 | 28.14 | 10.76 | 42.04 | 59.95 | 32.61 | 31.10 |

*Table 2.* Results of mixed training experiments on DOQA datasets

| Train Method | Gnenrate Method | F1 | Rouge-l | Bleu-1 | Bleu-2 | Bleu-3 | Bleu-4 | Dist-1 | Dist-2 | Dist-3 |
|---|---|---|---|---|---|---|---|---|---|---|
| | | | | | Qwen1.5-0.5B | | | | | |
| causal+Prefix-I | causal | 37.20 | 36.41 | 30.59 | 27.33 | 25.32 | 24.23 | 12.68 | 51.93 | 72.47 |
| causal+Prefix-I | Prefix-I | 33.10 | 32.40 | 20.12 | 17.62 | 15.87 | 14.95 | 14.33 | 57.09 | 76.87 |
| causal+causal | causal | 36.58 | 35.82 | 30.28 | 26.98 | 25.00 | 23.93 | 12.57 | 52.40 | 73.18 |
| | | | | | Qwen1.5-1.8B | | | | | |
| causal+Prefix-I | causal | 39.12 | 38.45 | 34.07 | 31.22 | 29.38 | 28.34 | 13.16 | 54.85 | 76.67 |
| causal+Prefix-I | Prefix-I | 37.71 | 37.08 | 30.64 | 27.89 | 26.06 | 25.05 | 13.39 | 55.42 | 75.92 |
| causal+causal | causal | 38.85 | 38.18 | 34.35 | 31.52 | 29.71 | 28.69 | 12.92 | 54.83 | 76.91 |
| | | | | | Qwen1.5-4B | | | | | |
| causal+Prefix-I | causal | 40.27 | 39.81 | 19.06 | 18.17 | 17.00 | 16.32 | 15.94 | 64.60 | 86.36 |
| causal+Prefix-I | Prefix-I | 35.60 | 35.27 | 9.32 | 9.01 | 8.24 | 7.81 | 16.90 | 69.16 | 87.46 |
| causal+causal | causal | 40.70 | 40.34 | 19.28 | 18.58 | 17.49 | 16.87 | 16.10 | 64.59 | 85.59 |

*Prefix-H*, the degree of bidirectional attention of the model is increasing. We show the experimental results in Figure 2. The experiment results on all tested datasets show that from *Cuaual* to *Prefix-H*, the model generation performance decreases with the increasing degree of bidirectional attention.

**Overfitting.** As show in Table 1. We used the DOQA dataset to fine-tune Qwen 1.5-0.5b and 1.5B, and we compared the effects of using Prefix-I with causal. The results show that on the DOQA data set, Prefix-I produces obvious overfitting and overlearns the unanswerable of the train data set.

**Hybrid mask training.** We conducted mixed training on the two methods causal and Preifx-I, and the experimental results are shown in Table 2. We were surprised to find that using causal+Prefix-I in both the 0.5B and 1.8B models

resulted in a slight increase in unidirectional generation performance, indicating some bidirectional advantage in understanding long text.

### 3.2. Average Preceding and Subsequent Dependence

To analyze the reason for the decline of productive ability caused by bidirectional attention, we propose an average subsequent and preceding dependence method that uses attention dependence to explain this phenomenon.

The specific explanation is shown in Figure 3. We first introduce the attention dependence of a single layer. For the attention weight $w = \text{softmax}(\mathbf{Q}\mathbf{K}^T) \in R^{b \times l \times l}$ (where $\mathbf{K}$ is the key, $\mathbf{Q}$ is the query, $b$ is head number and $l$ is token sequence length) of the $k$-th layer in the $n$ self-attention layers, we sum it to obtain the attention score $A_k \in R^{l \times l}$ between each pair of tokens. We then average the $A_k$ of $n$

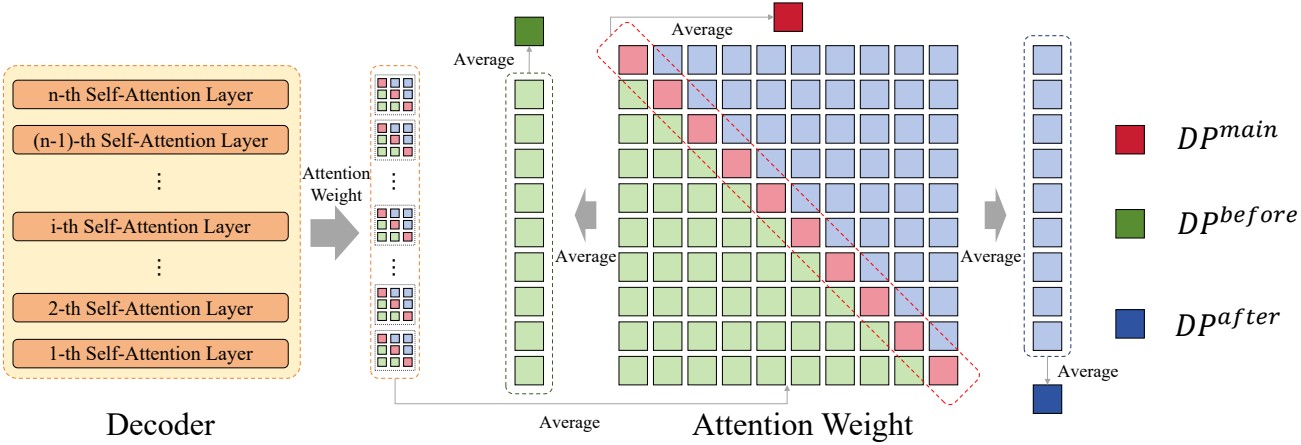

*Figure 3.* The calculation process of average before and after dependence.

self-attention layers to obtain the attention weight $A$ for the global model. We assume that $A_{i,j}$ represents the dependence of the $i$-th token on the $j$-th token, then the average dependence of the $i$-th token on the preceding $i-1$ token $DP_i^{before}$ and the average dependence of the following $l-i$ token $DP_i^{after}$ can be expressed as:

$$DP_i^{before} = \frac{1}{i-1} \sum_{j=1}^{i-1} A_{i,j},$$

$$DP_i^{after} = \frac{1}{l-i} \sum_{j=0}^{i} A_{i,j}. \tag{1}$$

$DP_i^{before}$ and $DP_i^{after}$ reflect the degree of dependence of the $i$-th token on the preceding and the following tokens respectively. In addition, it is easy to understand that $A_{i,i}$ represents the degree of dependence of the $i$-th token on itself, which we denote by $DP_i^{main}$.

In order to obtain the dependence of the full text on the preceding text, the following text and itself, we carry out an average operation on each dependence indicator for each token:

$$DP^{indicator} = \frac{1}{l'} \sum_{i=1}^{l'} DP_i^{indicator}, \tag{2}$$

$$indicator \in \{main, before, after\}.$$

It is worth noting that $l' = l$ when indicator is main while $l' = l-1$ when $indicator$ is $berfore$ or $after$. Because the calculating process of the $DP_i^{before}$ or $DP_i^{after}$ will not get $DP_1^{before}$ or $DP_l^{after}$, which can be easy to understand by Figure 3.

We conduct experiments on the average attention dependence metric with different training objectives, different token lengths, and different model sizes. The experiment setting can be found in Section 5.1. As shown in Table 3, after training with bi-directional contrastive learning, the model's subsequent dependence shows a significant increase. Through the discussion of different token lengths and different model scales, it can be considered that the source of this subsequent dependence is bidirectional training. Additionally, we observes that increasing the model scale and the number of tokens also leads to an increase in subsequent dependence. As the subsequent dependence increases, we can see a downward trend in performance on the MMLU dataset. This trend demonstrated remarkable consistency across the four model scales. This fully reflects the statistical association between subsequent dependence and model generation ability. We conduct the same experiment on LLAMA3-8B, and the results are shown in Table 3. This further validates our hypothesis that bidirectional training leads to the increase of subsequent dependence and thus damages the model generation ability.

## 4. UBMoE-LLM

Through pre-experiments and attention-dependence analysis, we draw the following conclusions: the performance degradation of text generation task caused by the join of bidirectional attention is mainly caused by the model's subsequent dependence and the FFN layer is least affected by the subsequent dependence during the bidirectional attention training process. Therefore, a novel Uni-Bi-directional Mixture of Expert Large Language Model, UBMoE-LLM, is proposed to optimize the FFN layer during the bidirectional training of the unidirectional generative model. The main architecture of our approach is shown in Figure 4. It joins the advantage of bidirectional attention on the basis of

*Table 3.* Dependence of four different scales of models in 0-2048 token lengths. *DP-M*, *DP-B*, and *DP-A* indicate $DP^{main}$, $DP^{before}$, and $DP^{after}$ respectively. The following tables take the same abbreviation. *Uni* and *Bi* indicate unidirectional and bidirectional respectively.

| Model | Attention | 0-256 | | | 256-512 | | | 512-1024 | | | 1024-2048 | | | MMLU |
|---|---|---|---|---|---|---|---|---|---|---|---|---|---|---|
| | | DP-M | DP-B | DP-A | DP-M | DP-B | DP-A | DP-M | DP-B | DP-A | DP-M | DP-B | DP-A | |
| Qwen1.5-0.5B-Chat | Uni | 4.83 | 63.48 | 31.69 | 3.48 | 59.57 | 36.95 | 3.01 | 56.62 | 40.37 | 2.61 | 51.45 | 45.94 | 32.5 |
| | Bi | 4.21 | 60.11 | **35.68** | 2.78 | 56.73 | **40.49** | 2.35 | 48.09 | **49.93** | 1.98 | 48.09 | **49.93** | 25.8 |
| Qwen1.5-1.8B-Chat | Uni | 4.44 | 67.23 | 28.33 | 3.21 | 64.15 | 32.64 | 2.82 | 61.51 | 35.68 | 2.47 | 55.32 | 44.21 | 44.3 |
| | Bi | 4.27 | 60.97 | **34.75** | 2.80 | 58.93 | **38.27** | 2.37 | 56.32 | **41.11** | 1.94 | 50.51 | **47.55** | 25.52 |
| Qwen1.5-4B-Chat | Uni | 4.28 | 68.3 | 27.43 | 3.30 | 66.66 | 30.04 | 2.93 | 65.16 | 31.9 | 2.68 | 61.62 | 35.69 | 54.2 |
| | Bi | 3.60 | 64.28 | **32.12** | 2.64 | 63.56 | **33.79** | 2.40 | 62.56 | **35.04** | 2.18 | 59.17 | **38.66** | 50.93 |
| Qwen1.5-7B-Chat | Uni | 4.48 | 71.10 | 24.42 | 3.30 | 69.34 | 27.36 | 2.95 | 67.81 | 29.24 | 2.80 | 63.65 | 33.55 | 60.2 |
| | Bi | 4.16 | 67.31 | **28.53** | 2.84 | 66.20 | **30.95** | 2.54 | 65.14 | **32.33** | 2.33 | 61.32 | **36.25** | 58.12 |
| Llama3-8B | Uni | 3.29 | 76.81 | 19.89 | 2.60 | 72.49 | 24.94 | 2.13 | 65.55 | 32.31 | 1.50 | 61.58 | 36.91 | 68.4 |
| | Bi | 3.05 | 75.42 | **21.53** | 2.51 | 70.31 | **27.18** | 2.02 | 61.42 | **36.56** | 1.48 | 56.48 | **41.72** | 65.4 |

maintaining the generative ability of unidirectional models, which can resist the hallucination problem faced by large models.

### 4.1. Uni-bi-directional Mixture-of-expert FFN Layer

To combine the context understanding ability of bidirectional attention with the text generation ability of unidirectional attention, we decided to adopt the Mixture of Experts (MoE) method to integrate bidirectional attention FFN layer into the unidirectional generative model. Specifically, We define the bidirectional attention FFN layer as bi-directional embedding expert **expert**$_{Bi}$, and the FFN layer of the backbone model is defined as uni-direction generation expert **expert**$_{Uni}$.

As shown in Figure 4 we implement the allocation of tokens through a gate control layer, activate only one expert at a time. Since the gate control layer does not get any training, we use a small amount of data to train the gate control layer while freezing other parameters. Following Jiang et al. (Jiang et al., 2023), we use cross entropy loss and gate control loss $L = L_{CE} + \lambda L_{Gating}$ to jointly train our model. $L_{CE}$ is the cross entropy loss and $L_{Gating}$ is the gating regularization loss. $\lambda$ is a hyperparameter that adjusts the weight between the cross-entropy loss and the gated regularization loss.

The Cross entropy loss is used to supervise the difference between the output of the learning model and the actual label. For a given input $x$ and target label $y$, the cross-entropy loss $L_{CE}$ is defined as:

$$L_{CE} = -\frac{1}{N} \sum_{i=1}^{N} \sum_{c=1}^{C} y_{ic} \log(p_{ic}), \quad (3)$$

where $N$ is the number of samples in the batch, $C$ is the number of categories, $y_{ic}$ is the actual label (one-hot coding) of the $i$ sample belonging to the $c$ category, and $p_{ic}$ is the

probability of the model predicting that the $i$ sample belongs to the $c$ category.

The gating regularization loss $L_{Gating}$ is to ensure uniform use of all experts. It uses negative entropy to encourage a uniform distribution of the gated network. It is specifically defined as:

$$L_{Gating} = -\frac{1}{N} \sum_{i=1}^{N} \sum_{j=1}^{M} g_{ij} \log(g_{ij}), \quad (4)$$

where $M$ is the number of experts and $g_{ij}$ is the probability that the $i$ th sample will have the $j$ th expert selected by the gating network.

### 4.2. Contrastive Learning based Bidirectional Expert Training

We use contrastive learning to train the model to obtain the FFN Layer containing the prior knowledge of bidirectional attention. For the input sequence $x = \{x_1, ..., x_t\}$, we simultaneously input $x$ with its similar positive cases $x^+$ and different negative cases $x^-$. After encoding and pooling of the model, we can obtain the embedding representation of the input sequence with its positive and negative cases $h$, $h^+$ and $h^-$. Since the model is the decoder-only model, we choose the vector corresponding to the token at the end of sentence ([EOS] token) position as the pooling operation. After that, we followed Gao et al. (Gao et al., 2021) 's comparative training objective for training:

$$loss = -\log \frac{e^{sim(h_i, h^+)/\tau}}{\sum_{j=1}^{N}(e^{sim(h_i, h_j^+)/\tau} + e^{sim(h_i, h_j^-)/\tau})}, \quad (5)$$

where $\tau$ is a temperature hyperparameter and $sim(h_1, h_2)$ is the cosine similarity $\frac{h_1^T h_2}{||h_1|| \cdot ||h_2||}$. By contrast learning

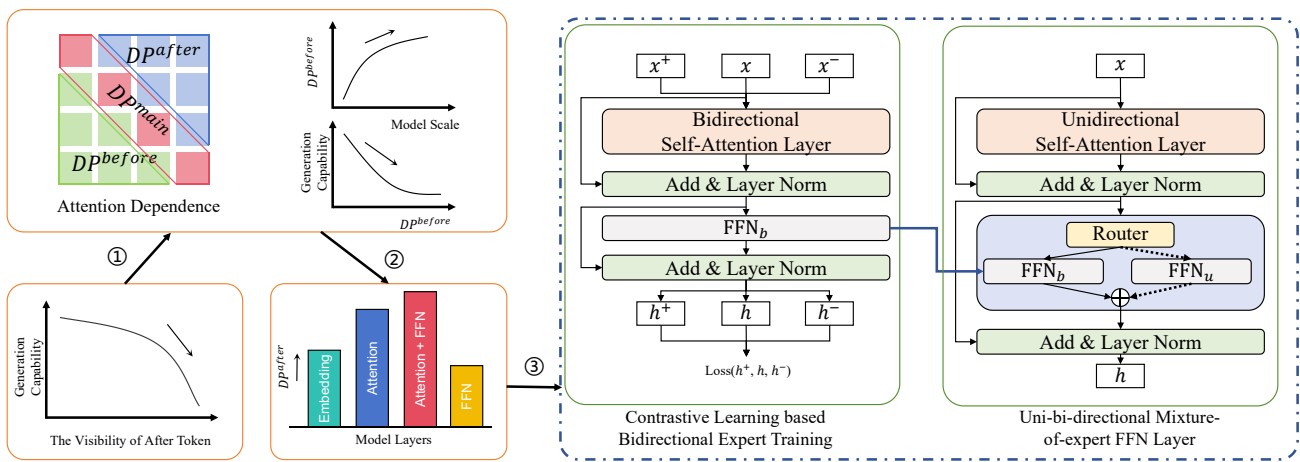

*Figure 4.* The model architecture and design ideas of UBMoE-LLM.

with positive and negative examples, the model can better grasp the bidirectional understanding ability of text. It is worth noting that the training process freezes the weights of other modules and trains only the FFN layer of the model. Therefore, we can obtain FFN layers based on bidirectional attention.

## 5. Experiments

### 5.1. Experiments Setup

**Datasets.** For the training of bidirectional contrast learning, we used the data collected by Li at al. (Li & Li, 2024a) to train our model. There were 480,862 training data in this dataset. Following the settings set by Muennighoff et al. (Muennighoff et al., 2024), we applied the model's default prompt template when training.

To test the performance of different fine-tuning layers on embedding tasks, we conduct experiments on ten text similarity (STS) tasks in MTEB (Muennighoff et al., 2022)

For the evaluation of subsequent dependence and the training for gate layer, we use tulu-v2-sft-mixture (Ivison et al., 2023), which is a multitask instruction dataset with a wide text interval, with 326,154 training data. We split it into four subsets based on the number of tokens recorded in the historical conversation, in other to dissolve the effect of token length on subsequent dependence. For the evaluation of all language models, we use the officially provided prompt template to process the input.

To evaluate the generative performance of UBMoE-LLM , we use the following three datasets: (1) MMLU: a dataset to evaluate AI language understanding across various subjects using a massive multitask framework (Hendrycks et al., 2021). (2) Winogrande: a large-scale dataset focused on common sense reasoning, designed to improve and evaluate

AI models on ambiguity resolution tasks (Sakaguchi et al., 2019). (3) TruthfulQA: a dataset designed to challenge AI models on their ability to generate truthful and factual responses in a question-answering format (Lin et al., 2022). During evaluation, the few-shot setting for MMLU is set to 5, using accuracy (acc) as the metric; for TruthfulQA, the few-shot setting is set to 0, using multiple-choice score (mc2) as the metric; for Winogrande, the few-shot setting is set to 5, also using accuracy (acc) as the metric.

**Model Scales.** To ablate the impact of model scale on dependence, we evaluate models of various scales. To minimize the influence of other factors, we select the Qwen1.5-Chat series models, ranging from 0.5B to 7B scales. These models have identical architecture, tokenizer, and instruction templates.

**Baselines.** For the embedding tasks, previous work has demonstrated that enabling bidirectional attention in unidirectional language models and training them using contrastive learning methods can effectively improve the embedding capability of the models. We followed this approach to train our model. We conducted the following comparative experiments: (1) ATT/FFN/EMB: updating only the parameters of the self-attention layer, FFN layer, or embedding layer, (2) EMB+ATT/FFN: updating the embedding layer while also updating the self-attention layer or FFN layer, (3) FFN+ATT: updating only the FFN layer and the self-attention layer, consistent with the parameter update settings in LLM2VEC (BehnamGhader et al., 2024), (4) ALL: updating all parameters.

For the dependence evaluation, the uni-bi-directional mixture-of-expert large language model and bidirectional contrastive learning, we use the backbone model as the baseline.

*Table 4.* Experimental results of word embedding ablation. *ATT*, *FFN*, and *EMB* indicate freezing other parameters for bidirectional modeling training on the Attention Layer, FFN Layer, and Embedding Layer respectively. The following tables take the same abbreviation. *ALL* Indicates the full-parameter bidirectional modeling trained model.

| Method | BIOSSES | SICK-R | STS12 | STS13 | STS14 | STS15 | STS16 | STS17 | STS22 | STSB | Average |
|---|---|---|---|---|---|---|---|---|---|---|---|
| **Qwen1.5-0.5B-Chat** | | | | | | | | | | | |
| Base | 31.56 | 53.56 | 24.96 | 35.86 | 22.54 | 21.65 | 52.30 | 51.50 | 12.91 | 42.76 | 34.96 |
| FFN | 54.01 | 69.73 | 72.41 | 68.16 | 65.86 | 75.71 | 70.89 | 77.38 | 42.82 | 72.61 | 66.96 |
| ATT | 55.07 | 68.30 | 70.5 | **73.67** | 68.51 | **77.41** | 73.56 | **79.19** | 43.57 | 73.15 | 68.29 |
| EMB | 27.77 | 54.32 | 34.67 | 30.18 | 24.49 | 23.09 | 52.19 | 59.94 | 20.52 | 48.02 | 37.52 |
| FFN+ATT | 49.17 | 66.73 | 71.85 | 67.08 | 65.09 | 75.82 | 72.08 | 76.42 | **48.59** | 73.49 | 66.63 |
| FFN+EMB | **58.38** | **69.89** | **72.69** | 71.18 | **68.86** | 77.05 | 73.04 | 77.62 | 41.44 | **73.61** | **68.38** |
| ATT+EMB | 58.32 | 69.02 | 70.43 | 71.55 | 67.52 | 76.20 | **74.27** | 79.02 | 41.66 | 73.55 | 68.15 |
| ALL | 51.98 | 68.10 | 72.06 | 68.38 | 65.52 | 74.66 | 72.38 | 75.05 | 49.19 | 71.98 | 66.93 |
| **Qwen1.5-1.8B-Chat** | | | | | | | | | | | |
| Base | 43.87 | 56.82 | 48.15 | 65.80 | 52.57 | 62.08 | 63.35 | 72.26 | 33.75 | 56.84 | 55.55 |
| FFN | **64.95** | 67.45 | **73.85** | 71.61 | 70.11 | **78.25** | **73.33** | 77.48 | 48.59 | 75.50 | 70.11 |
| ATT | 58.42 | 66.24 | 72.09 | 74.28 | 70.97 | 77.73 | 72.67 | 75.69 | 47.11 | 75.15 | 69.04 |
| EMB | 54.90 | **71.66** | 63.23 | 65.82 | 61.56 | 72.77 | 66.72 | 77.57 | 23.74 | 70.71 | 62.87 |
| FFN+ATT | 63.99 | 67.03 | 72.69 | 73.39 | 71.08 | 78.03 | 73.03 | 76.32 | 46.64 | 75.26 | 69.75 |
| FFN+EMB | 66.3 | 67.6 | 72.58 | 72.55 | 70.57 | 78.11 | 73.13 | 76.82 | **50.7** | 75.37 | **70.37** |
| ATT+EMB | 58.28 | 65.86 | 72.22 | 73.41 | 70.69 | 77.72 | 72.55 | 76.15 | 45.42 | 75.70 | 68.80 |
| ALL | 61.80 | 66.48 | 72.95 | **74.33** | **71.23** | 77.93 | 72.87 | **77.85** | 48.10 | **75.71** | 69.93 |
| **Qwen1.5-4B-Chat** | | | | | | | | | | | |
| Base | 31.00 | 53.68 | 24.04 | 40.75 | 24.55 | 25.59 | 52.40 | 49.35 | 10.29 | 41.34 | 35.30 |
| FFN | **64.97** | 63.99 | 72.30 | 75.62 | 71.97 | **78.08** | **76.87** | 69.53 | 54.93 | 76.51 | 70.48 |
| ATT | 64.96 | **64.74** | 71.56 | **77.54** | **73.9** | 77.85 | 75.54 | **73.46** | **57.13** | **77.35** | **71.40** |
| EMB | 4.44 | 35.48 | 27.78 | 17.13 | 15.62 | 22.27 | 28.05 | 37.34 | 10.63 | 25.60 | 22.43 |
| FFN+ATT | 69.17 | 63.73 | **71.86** | 76.59 | 73.17 | 78.00 | 76.11 | 71.66 | 56.44 | 76.43 | 71.32 |
| FFN+EMB | 63.03 | 61.16 | 51.78 | 35.94 | 45.57 | 61.32 | 51.36 | 65.33 | 41.35 | 54.72 | 53.16 |
| ATT+EMB | 51.16 | 60.99 | 53.53 | 41.90 | 45.70 | 59.55 | 52.68 | 63.91 | 39.45 | 54.60 | 52.35 |
| ALL | 62.16 | 59.76 | 51.43 | 39.6 | 47.14 | 59.48 | 51.01 | 67.05 | 39.53 | 55.97 | 53.31 |
| **Qwen1.5-7B-Chat** | | | | | | | | | | | |
| Base | 23.79 | 56.77 | 11.94 | 40.85 | 22.45 | 26.43 | 52.39 | 43.85 | 12.21 | 29.59 | 32.03 |
| FFN | **66.20** | 64.73 | **73.68** | 79.74 | 75.08 | 79.04 | 73.87 | 73.21 | 49.36 | **78.55** | 71.35 |
| ATT | 63.87 | 65.12 | 73.52 | **80.25** | **75.44** | **80.26** | 77.62 | 73.11 | **58.24** | 78.23 | **72.57** |
| EMB | 29.94 | 53.82 | 37.90 | 25.33 | 26.95 | 32.22 | 42.70 | 48.39 | 7.42 | 32.46 | 33.71 |
| FFN+ATT | 65.52 | **65.17** | 73.20 | 79.80 | 74.61 | 79.59 | **77.73** | **71.45** | 57.27 | 77.92 | 72.23 |
| FFN+EMB | 60.69 | 62.18 | 51.06 | 36.70 | 45.57 | 60.39 | 52.45 | 66.65 | 32.52 | 54.62 | 52.28 |
| ATT+EMB | 61.55 | 60.31 | 51.87 | 31.64 | 43.92 | 60.25 | 51.70 | 68.81 | 31.00 | 55.94 | 51.70 |
| ALL | 52.05 | 60.04 | 50.72 | 36.14 | 46.03 | 61.13 | 52.93 | 65.97 | 31.41 | 55.81 | 51.23 |

**Model Settings.** For bidirectional contrasted learning training, we conduct full fine-tuning for the 0.5B and 1.8B models, and use LoRA (Hu et al., 2021) for parameter adjustments on the 4B and 7B models. We fix the learning rate at 0.00003 but vary the batch size and gradient accumulation steps according to the model parameters, ensuring their product remains 32, and apply the same hyperparameter settings within the same scale models. For LoRA, we set $r = 16$, $\alpha = 32$, LoRA dropout to 0.05, and enable reLoRA (Lialin et al., 2023).

For the training of UBMoE-LLM, we only sample data with token counts less then 512. The batch size is set to 1, gradient accumulation is set to 32, and the learning rate is set at 0.0003, with training limited to 1000 steps. During the training process, we freeze parameter updates except for those of the gating layer. We trained Qwen1.5-Chat with sizes 0.5B, 1.8B, and 4B on a GeForce RTX 4090. For Qwen1.5-7B-Chat and Llama3-8B, we train them on a NVIDIA L20. For the evaluation of UBMoE-LLM, we use the scripts provided by Language Model Evaluation Harness for evaluation (Gao et al., 2023).

### 5.2. Analysis

**Bidirectional embedding expert** The embedding tasks experiment results are shown in Table 4. In all sizes of models, although the optimal results of fine-tuning different layers are different, merely fine-tuning the FFN or attention layers has achieved robust performance in the text similar-

*Table 5.* Contrastive learning subsequent dependence ablation.

| Model | Trained Module | 0-256 | | | 256-512 | | | 512-1024 | | | 1024-2048 | | |
|---|---|---|---|---|---|---|---|---|---|---|---|---|---|
| | | DP-M | DP-B | DP-A | DP-M | DP-B | DP-A | DP-M | DP-B | DP-A | DP-M | DP-B | DP-A |
| Qwen1.5-0.5B-Chat | FFN | 3.86 | 61.74 | **34.40** | 2.58 | 57.87 | **39.55** | 2.22 | 54.73 | **43.05** | 1.92 | 49.56 | **48.52** |
| | ATT | 4.33 | 59.28 | 36.39 | 2.82 | 56.44 | 40.74 | 2.39 | 53.49 | 44.12 | 2.00 | 48.53 | 49.48 |
| Qwen1.5-1.8B-Chat | FFN | 3.82 | 62.81 | **33.37** | 2.65 | 60.79 | **36.56** | 2.34 | 58.85 | **38.81** | 2.04 | 52.72 | **45.24** |
| | ATT | 4.09 | 62.49 | 33.41 | 2.52 | 60.68 | 36.79 | 2.10 | 58.41 | 39.48 | 1.69 | 52.98 | 45.33 |
| Qwen1.5-4B-Chat | FFN | 3.64 | 65.60 | **30.76** | 2.71 | 64.31 | **32.98** | 2.43 | 63.14 | **33.43** | 2.19 | 59.62 | **38.18** |
| | ATT | 3.62 | 62.52 | 33.86 | 2.63 | 63.09 | 35.28 | 2.38 | 61.17 | 36.44 | 2.17 | 57.91 | 39.92 |
| Qwen1.5-7B-Chat | FFN | 4.05 | 69.64 | **26.32** | 2.87 | 68.16 | **28.97** | 2.57 | 67.17 | **30.25** | 2.41 | 63.44 | **34.15** |
| | ATT | 3.98 | 65.39 | 30.63 | 2.63 | 64.81 | 32.56 | 2.29 | 63.67 | 34.04 | 2.06 | 59.67 | 38.29 |

*Table 6.* Experiment results of UBMoE-LLM and base model with different model scales on the generation tasks. All results are evaluated by us.

| Model | Model Scale | MMLU | Winogrande | Truthfulqa | Average |
|---|---|---|---|---|---|
| Qwen1.5-Chat | 0.5B | 32.5 | 53.0 | 43.0 | 42.8 |
| UBMoE-LLM | | 28.1 | 54.0 | 43.9 | 42.0 |
| Qwen1.5-Chat | 1.8B | 44.3 | 59.7 | 40.5 | 48.2 |
| UBMoE-LLM | | 45.2 | 58.5 | 40.9 | 48.2 |
| Qwen1.5-Chat | 4B | 54.2 | 66.0 | 44.5 | 54.9 |
| UBMoE-LLM | | 51.9 | 64.7 | 47.6 | 54.7 |
| Qwen1.5-Chat | 7B | 60.2 | 67.4 | 53.7 | 60.4 |
| UBMoE-LLM | | 59.6 | 66.0 | 55.4 | 60.2 |

ity computation tasks across all scales. We evaluated the dependence distribution of models trained via bidirectional contrastive learning for these two components and token evaluations as show in Table 5. The results indicate that solely fine-tuning the FFN consistently yielded the lowest subsequent dependence across various model scales. It is shown that only fine-tuning the FFN has the lowest impact on the dependence of the model. This is primarily because, in the initial layers where the attention layer is not updated, the attention distribution during the bidirectional training phase is exactly the same as the unidirectional generation. In this context, the FFN layer mainly learns how to process hidden states that contain backward token information under the same attention distribution. Updating the ATT layer, however, would disrupt this consistency in distribution. This implicit separation of the two modeling capabilities reduces the performance degradation of the FFN in unidirectional modeling during training.

**UBMoE-LLM** The generative performance of UBMoE-LLM is illustrated in Table 6, where our method consistently achieves improvements on the TruthfulQA dataset while maintaining robust generative capabilities. At the same time, due to only learning token allocation capabilities during the training phase, degradation in some generation tasks is inevitable. This is because the bidirectional experts, after training on word embeddings, did not undergo instruction alignment, leading to a decline in their generative abilities. Notably, our approach manages to enhance overall generative performance at the 1.8B scale, which suggests the benefits of incorporating bidirectional experts—enhancing the model's resistance to hallucinations and improving its language comprehension capabilities and the richness of semantic modeling. More experiment results can be found in Appendix B.

## 6. Conclusion

To explore whether bidirectional training of unidirectional models would enhance subsequent dependence, we reinterpreted attention weights as a form of dependency, dividing it into preceding dependency, subsequent dependence, and self/main dependency. We have provided the formulas for calculating these dependencies and tested our hypothesis on four models ranging from 0.5B to 7B — bidirectional training does increase subsequent dependence. Additionally, we performed an ablation study on the bidirectional contrastive learning training part, revealing that training only the FFN layers results in robust embedding effects and a lower increase in subsequent dependence. We used the fine-tuned FFN layers as bidirectional embedding experts, paired with the unidirectional generative experts of the main model. By adding a gating mechanism, we constructed a hybrid expert language model that effectively utilizes the illusion resistance brought by bidirectionality and the robust generative capabilities of unidirectionality. This model showed consistent improvements on the realness benchmark dataset TruthfulQA.

## Acknowledgments

This work was supported by the National Natural Science Foundation of China (No. 62306216), the Technology Innovation Program of Hubei Province (No. 2024BAB043) and the National Social Science Fund of China (No. 24&ZD186).

## Impact Statement

This paper presents work whose goal is to advance the field of Machine Learning. There are many potential societal consequences of our work, none which we feel must be specifically highlighted here.

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

## A. Effect of bidirectional attention model

In Table 7, we follow Li et al. (Li et al., 2023b) 's work to train a bidirectional attention model BatGPT and test it on the CMMLU (Li et al., 2023a) dataset. We can find that BatGPT with bi-directional attention has excellent performance, fully demonstrating the benefits of bi-directional attention.

*Table 7.* Performance comparison of BATGPT and other Chinese-oriented large language models on the CMMLU benchmark. Results are presented as average accuracy within each categories based on a five-shot experiment setting.

| Model | STEM | Humanities | Social Science | Other | China-specific | **Average** |
|---|---|---|---|---|---|---|
| Random | 25.00 | 25.00 | 25.00 | 25.00 | 25.00 | 25.00 |
| MOSS-SFT-16B | 27.23 | 30.41 | 28.84 | 32.56 | 28.68 | 29.57 |
| Chinese-LLaMA-13B | 27.12 | 33.18 | 34.87 | 35.10 | 32.97 | 32.63 |
| Chinese-GLM-10B | 25.49 | 27.05 | 27.42 | 29.21 | 28.05 | 27.26 |
| Chinese-LLaMA-7B | 25.79 | 27.45 | 26.35 | 26.06 | 25.45 | 26.36 |
| ChatGLM-6B | 32.35 | 39.22 | 39.65 | 38.62 | 37.70 | 37.48 |
| BATGPT | **33.49** | 35.38 | 36.31 | **42.14** | 37.00 | 36.72 |

## B. Further Study

**Performance on more base models.** We analyzed the relationship between generation performance and attention-based forward or subsequent dependence on other models. The experimental results are shown in Table 8. As the level of subsequent dependence increased, the generation performance of the model declined. In contrast, our method reduced the subsequent dependence introduced by incorporating bidirectional experts while improving generation performance. This indicates a correlation between the distribution of attention weights and the model's generation performance.

| Model | DPB | DPA | MMLU |
|---|---|---|---|
| Qwen2.5-0.5B-Instruct | 55.68 | 39.75 | 46.9 |
| UBMoE-LLM | 54.68 | 38.91 | 43.85 |
| FNN-Bi | 54.40 | 41.40 | 35.54 |
| Qwen2.5-1.5B-Instruct | 58.27 | 36.57 | 60.21 |
| UBMoE-LLM | 58.08 | 36.80 | 59.41 |
| FNN-Bi | 57.00 | 38.13 | 54.35 |
| Qwen2.5-7B-Instruct | 60.94 | 35.07 | 73.86 |
| UBMoE-LLM | 60.37 | 35.61 | 70.86 |
| FNN-Bi | 46.17 | 49.05 | 21.28 |

*Table 8.* Dependence and Performance on Qwen 2.5.

Besides, we add MNTP-LLM as a baseline, which is trained using the Wikipedia dataset through the MNTP method. As shown in Table 9, the experimental results demonstrate that UBMoE-LLM still exhibits strong performance. we also test the generation ability on models beyond 7B parameters. As shown in Table 10, the experimental results of the 14B model remain consistent with the previous conclusions.

| Model | MMLU | Winogrande | Truthfulqa | Avg |
|---|---|---|---|---|
| Qwen2.5-0.5B-Instruct | 46.90 | 55.60 | 41.86 | 48.12 |
| UBMoE-LLM | 43.85 | 56.94 | 43.02 | 47.94 |
| MNTP-LLM | 41.33 | 54.40 | 41.65 | 45.79 |
| Qwen2.5-7B-Instruct | 73.86 | 73.60 | 64.72 | 70.72 |
| UBMoE-LLM | 70.86 | 75.42 | 62.65 | 69.64 |
| MNTP-LLM | 70.06 | 74.21 | 62.54 | 68.93 |

*Table 9.* Performance on generation tasks.

| Model | MMLU | Winogrande | Truthfulqa | Avg |
|---|---|---|---|---|
| Qwen2.5-14B-Instruct | 79.54 | 78.74 | 67.51 | 75.26 |
| UBMoE-LLM | 78.44 | 77.83 | 68.01 | 74.76 |

*Table 10.* Performance on 14B model.

**Comparison of computational overhead.** UBMoE activates only one expert for each token, adding computational overhead during inference solely for token routing. Compared to the computational cost of the causal language model itself, this additional overhead is minimal. As shown in Table 11, our approach does not result in significant additional computational overhead.

| Model | GFLOPs |
|---|---|
| Qwen2.5-0.5B-Instruct | 505.82 |
| UBMoE-LLM (0.5B) | 505.86 |
| Qwen2.5-1.5B-Instruct | 1,580.62 |
| UBMoE-LLM (1.5B) | 1,580.70 |
| Qwen2.5-7B-Instruct | 7,239.97 |
| UBMoE-LLM (7B) | 7,240.18 |

*Table 11.* Comparison of computational overhead.

**Performance on reasoning tasks.** To evaluate the reasoning ability of UBMoE-LLM, we evaluate the performance of UBMoE-LLM on a physics problem benchmark and a mathematics problem benchmark. As shown in Table 12, the experimental results indicate that UBMoE-LLM still demonstrates performance comparable to that of a causal language model.

| Model | Math | PIQA | Avg |
|---|---|---|---|
| Qwen2.5-0.5B-Instruct | 32.5 | 70.62 | 51.56 |
| UBMoE-LLM | 32.67 | 69.85 | 51.26 |
| Qwen2.5-1.5B-Instruct | 41.07 | 76.22 | 58.64 |
| UBMoE-LLM | 42.68 | 76.36 | 59.52 |
| Qwen2.5-7B-Instruct | 54.51 | 79.49 | 67.00 |
| UBMoE-LLM | 53.91 | 79.04 | 66.48 |

*Table 12.* Performance on reasoning tasks.

**Ablation study of gate control layer.** To test the effectiveness of gate control layer, we carry out ablation experiments. As shown in Table 13, we conduct experiments on the STS task. The gate control layer effectively enhances the ability to embed tasks.

| Model | Method | Avg |
|---|---|---|
| Qwen2.5-0.5B-Instruct | FNN | 81.63 |
| Qwen2.5-0.5B-Instruct | FNN w/o gate | 80.14 |
| Qwen2.5-1.5B-Instruct | FNN | 80.15 |
| Qwen2.5-1.5B-Instruct | FNN w/o gate | 78.49 |
| Qwen2.5-7B-Instruct | FNN | 81.79 |
| Qwen2.5-7B-Instruct | FNN w/o gate | 80.39 |

*Table 13.* Ablation study of gate control layer.

