# OpenReview forum: "What Limits Bidirectional Model's Generative Capabilities? A Uni-Bi-Directional Mixture-of-Expert Method For Bidirectional Fine-tuning"
_ICML.cc/2025/Conference — ICML 2025 poster_

### Official Review · Reviewer_39Yf · 2025-02-25

**Overall Recommendation:** 5

**Summary:**

This paper explores the impact of bidirectional fine-tuning on the performance of unidirectional language models, particularly focusing on the decline in generative ability caused by bidirectional fine-tuning. The authors attribute this decline to subsequent dependence and perform an in-depth analysis to support this explanation. A key finding is that fine-tuning the Feed-Forward Network (FFN) layer has the least impact on generative performance. To address the trade-off between generative and embedding tasks, the paper introduces UBMoE-LLM, a novel model that leverages the Mixture-of-Experts (MoE) method. UBMoE-LLM integrates both the original FFN layer and the bidirectionally fine-tuned FFN layer to preserve generative capabilities while improving performance on embedding tasks. Experimental results demonstrate that UBMoE-LLM achieves good performance, showcasing its potential to balance efficiency and effectiveness in practical applications. This work has significant implications for advancing the adaptability of pre-trained models across diverse downstream tasks.

**Claims And Evidence:**

Yes, the claims made in the submission are largely supported by evidence provided in the paper, though the strength of the evidence may vary depending on the specific claim.

**Essential References Not Discussed:**

Based on the current analysis, the references cited in the paper appear to cover the core prior work in the field and provide sufficient background to support the key contributions of the study.

**Experimental Designs Or Analyses:**

The paper evaluates UBMoE-LLM on both generative and embedding tasks, comparing it to baseline models and demonstrating improvements. The overall experimental setup is reasonable, but more types of foundation models can be further evaluated.

**Methods And Evaluation Criteria:**

The proposed UBMoE-LLM are well-suited to the bidirectional finetune degeneration problem. However, the generality of the findings could be enhanced by broadening the evaluation to include more baselines, datasets, and task types.

**Other Comments Or Suggestions:**

There are also some minor improvements.
1. Test on More Models: The authors have only tested the Qwen 1.5 series of models. The experiments on more models are yet to be conducted.

2. Test on Recent Models: It is beneficial to conduct experiments on recent LLMs.

3. Experimental Details: Several important experimental details are missing or not properly explained.

    a) The experimental setup for Figure 2 is not provided in the main text. The authors need to clarify the specific LLMs used in the experiments, as this information is not available in Appendix C.

    b) The value of λ (lambda) used in the loss function is not specified. The authors need to explicitly provide this information.

    c) In Line 258, the authors mention using a "small amount of data" to train the gating layer, but they do not define what this dataset consists of. Further clarification is needed.

4. Typos: Some typos remain to be corrected.

    a) The word "FNN" should be corrected to "FFN".

    b) The word "Casual" should be corrected to "Causal".

**Other Strengths And Weaknesses:**

1. This paper demonstrates an interesting exploration in attention. The authors propose the concepts of preceding and subsequent dependence to explain the relationships between tokens in the attention layer. The experiments on attention show the dependence of attention when bidirectionally fine- tuning different layers.

2. The authors have validated the method at four different scales, and the wide experimental scope provides insights into the relationship between performance and parameter count, offering guidance for scaling up to larger models.

3. The authors combine the bidirectionally fine-tuned FFN layer through the MoE method to achieve the model's embedding ability while retaining its generative ability. The proposed method is innovative and well-grounded. This series of models holds potential application value.

4. Although there are some minor issues in wording, this paper provides a well-justified and impactful solution to the bidirectional fine-tuning of LLMs that deserves to be published.

**Questions For Authors:**

It seems that FFN+ATT, as well as FFN+EMB perform better on the model size of 0.5B in Figure 2. I would like to understand why this is the case.

**Relation To Broader Scientific Literature:**

The key contributions of the paper—identifying subsequent dependence as a cause of generative performance degradation, proposing UBMoE-LLM to balance task performance, and demonstrating the importance of FFN layer fine-tuning—are well-grounded in the broader scientific literature. The paper builds on prior work in bidirectional vs. unidirectional models, MoE methods, and layerwise fine-tuning while advancing the field by addressing a specific trade-off between generative and embedding tasks. These contributions represent a meaningful extension of existing research, with the potential to influence future work in fine-tuning paradigms and multi-task NLP models.

**Theoretical Claims:**

Yes, the paper provides a theoretical explanation for subsequent dependence, which is likely grounded in how bidirectional fine-tuning alters the model’s internal representations or dependencies that are crucial for unidirectional generation.

---

> ### Author Rebuttal · Authors · 2025-04-01
>
> > Weakness 1&2: Test on More Models: The authors have only tested the Qwen 1.5 series of models. The experiments on more models are yet to be conducted. Test on Recent Models: It is beneficial to conduct experiments on recent LLMs.
>
> Thank you for your advice. We evaluate the effectiveness of our proposed method on the recently released Qwen2.5 series, and UBMoE still achieved performance comparable to that of causal language models.
>
> | Model                 | MMLU  | Winogrande | Truthfulqa | Avg   |
> | --------------------- | ----- | ---------- | ---------- | ----- |
> | Qwen2.5-0.5B-Instruct | 46.90 | 55.60      | 41.86      | 48.12 |
> | UBMoE-LLM             | 43.85 | 56.94      | 43.02      | 47.94 |
> | Qwen2.5-1.5B-Instruct | 60.21 | 64.48      | 46.58      | 57.09 |
> | UBMoE-LLM             | 59.41 | 63.95      | 49.26      | 57.54 |
> | Qwen2.5-7B-Instruct   | 73.86 | 73.60      | 64.72      | 70.72 |
> | UBMoE-LLM             | 70.86 | 75.42      | 62.65      | 69.64 |
> | Qwen2.5-14B-Instruct  | 79.54 | 78.74      | 67.51      | 75.26 |
> | UBMoE-LLM             | 78.44 | 77.83      | 68.01      | 74.76 |
>
>
> > Weakness 3: Experimental Details: Several important experimental details are missing or not properly explained.
>
> We provide the experimental setup for Figure 2 in Appendix B. For the model, we use Qwen1.5-0.5B.
>
> We set the value of λ (lambda) to 0.001.
>
> We provide detailed data sources in Section 5.1. The data used to train the gating layer is sampled from the Tulu-v2-SFT mixture, with only 32,000 samples used for training. These details will be added to Section 5.1 in the revised manuscript.
>
> > Question 1: It seems that FFN+ATT, as well as FFN+EMB perform better on the model size of 0.5B in Figure 2. I would like to understand why this is the case.
>
> For LLMs with fewer parameters, fine-tuning both the FFN and attention layers can yield modest performance gains due to the limited model capacity, as shown in our results in Table 2. However, as the model size increases, this performance gap narrows significantly.
>
> > Typos
>
> Thank you for your thorough review. We will address this and carefully check our paper in a next revision.
>
> > Summary
>
> Thank you for your careful reading and suggestions. We will improve our work according to your suggestion. We hope these additional analyses address your concerns and improve the manuscript's rigor.

---

> > ### Comment · Reviewer_39Yf · 2025-04-05
> >
> > I appreciate the answers given and change my score to 5.

---

> > > ### Author Response · Authors · 2025-04-05
> > >
> > > Thank you for your reply and encouragement to us.

---

### Official Review · Reviewer_y1xr · 2025-03-10

**Overall Recommendation:** 3

**Summary:**

There exists a common belief that causal language models (i.e., unidirectional models) perform better in generation tasks while bidirectional models perform better in embedding tasks. However, bidirectional finetuning of unidirectional models usually leads to significantly inferior generation performance, which makes it difficult to obtain a model that excels in both tasks. This paper analyzes the performance degradation from a new perspective, i.e., they analyze the attention scores and observe that bidirectional finetuning enhances subsequence dependence. Furthermore, they find that training only the FNN layers results in a lower increase in subsequent dependence. Based on that, they propose UBMoE-LLM, a new bidirectional finetuning paradigm that combines the original FNN layer of the unidirectional model with the bidirectional FNN layer trained by unsupervised contrastive learning through MoE and exhibits impressive performance in generation and embedding tasks.

**Claims And Evidence:**

The authors claim that the inferior performance of bidirectional finetuning is due to the increased subsequence dependence. The motivation is easy to follow. However, although the results in Table 1 are interesting, I am afraid they are not enough to verify the relationship between subsequence dependence and generation performance. I think more verification experiments will make this conclusion more convincing.

**Essential References Not Discussed:**

Most of the essential related works are discussed in this paper.

**Experimental Designs Or Analyses:**

The results in Tables 2, 3, 4 demonstrate the performance of bidirectional training in both generation and embeddings tasks. However, to show the effectiveness of UBMoE-LLM, I believe additional comparisons with other bidirectional training methods and bidirectional models are necessary. The current results are not enough to support the claims proposed by the authors.

**Methods And Evaluation Criteria:**

The solutions cooperate well with the empirical findings (tuning FFN layers results in a lower increase in subsequent dependence). However, I believe additional ablation studies would be helpful to analyze the role of the gate control layer.

**Other Comments Or Suggestions:**

N/A

**Other Strengths And Weaknesses:**

1. Obtaining a model that performs well in both generation and embeddings tasks is a significant topic and this paper provides novel insights.
2. The paper is organized well and easy to follow.

**Questions For Authors:**

1. This paper focuses on bidirectional finetuning to enhance the performance of causal language models. I wonder whether it is possible to achieve comparable performance on embeddings tasks compared to bidirectional pretraining tasks.
2. I note that UBMoE-LLM improves the generation performance on TruthfulQA. Is it possible to provide an additional explanation for the improvements?

**Relation To Broader Scientific Literature:**

N/A

**Theoretical Claims:**

I do not find theoretical analysis in this paper.

---

> ### Author Rebuttal · Authors · 2025-04-01
>
> >  Weakness 1: The authors claim that the inferior performance ... this conclusion more convincing.
>
> As we replied to Reviewer bS72, Table 1 shows that there is a consistent trend between subsequent dependence and the general capability of the model. This indicates a correlation between subsequent dependence and the model’s general performance, suggesting a potential causal relationship.
>
> Besides, we analyzed the relationship between generation performance and attention-based forward or subsequent dependence. The experimental results are shown in the table below. As the level of subsequent dependence increased, the generation performance of the model declined. In contrast, our method reduced the subsequent dependence introduced by incorporating bidirectional experts while improving generation performance. This indicates a correlation between the distribution of attention weights and the model's generation performance.
>
> |         Model         |  DPB  |  DPA  | MMLU  |
> | :-------------------: | :---: | :---: | :---: |
> | Qwen2.5-0.5B-Instruct | 55.68 | 39.75 | 46.9  |
> |       UBMoE-LLM       | 54.68 | 38.91 | 43.85 |
> |        FNN-Bi         | 54.40 | 41.40 | 35.54 |
> | Qwen2.5-1.5B-Instruct | 58.27 | 36.57 | 60.21 |
> |       UBMoE-LLM       | 58.08 | 36.80 | 59.41 |
> |        FNN-Bi         | 57.00 | 38.13 | 54.35 |
> |  Qwen2.5-7B-Instruct  | 60.94 | 35.07 | 73.86 |
> |       UBMoE-LLM       | 60.37 | 35.61 | 70.86 |
> |        FNN-Bi         | 46.17 | 49.05 | 21.28 |
>
> > Weakness 2: The results in Tables 2, 3, 4 demonstrate ... current results are not enough to support the claims proposed by the authors.
>
> Thank you for your advice. We added MNTP-LLM as a baseline, which is trained using the Wikipedia dataset through the MNTP method. As shown in the table below, the experimental results demonstrate that UBMoE-LLM still exhibits strong performance.
>
> |         Model         | MMLU  | Winogrande | Truthfulqa |  Avg  |
> | :-------------------: | :---: | :--------: | :--------: | :---: |
> | Qwen2.5-0.5B-Instruct | 46.90 |   55.60    |   41.86    | 48.12 |
> |       UBMoE-LLM       | 43.85 |   56.94    |   43.02    | 47.94 |
> |       MNTP-LLM        | 41.33 |   54.40    |   41.65    | 45.79 |
>
> |        Model        | MMLU  | Winogrande | Truthfulqa |  Avg  |
> | :-----------------: | :---: | :--------: | :--------: | :---: |
> | Qwen2.5-7B-Instruct | 73.86 |    73.6    |   64.72    | 70.72 |
> |      UBMoE-LLM      | 70.86 |   75.42    |   62.65    | 69.64 |
> |      MNTP-LLM       | 70.06 |   74.21    |   62.54    | 68.93 |
>
> > Weakness 3: However, I believe additional ablation studies would be helpful to analyze the role of the gate control layer.
>
> Thank you for your helpful advice. As shown in the table below, we conducted experiments on the STS task. The gate control layer effectively enhances the ability to embed tasks.
>
> | Model                 | Method       | Avg   |
> | --------------------- | ------------ | ----- |
> | Qwen2.5-0.5B-Instruct | FNN          | 81.63 |
> | Qwen2.5-0.5B-Instruct | FNN w/o gate | 80.14 |
> | Qwen2.5-1.5B-Instruct | FNN          | 80.15 |
> | Qwen2.5-1.5B-Instruct | FNN w/o gate | 78.49 |
> | Qwen2.5-7B-Instruct   | FNN          | 81.79 |
> | Qwen2.5-7B-Instruct   | FNN w/o gate | 80.39 |
>
> > Question 1: This paper focuses on bidirectional finetuning ... embeddings tasks compared to bidirectional pretraining tasks.
>
> We appreciate the reviewer’s suggestion. However, it is well known that there is currently a lack of proper similar scale bidirectional LLMs to serve as a baseline. We are trying to train to the bidirectional model for testing.
>
> > Question 2: I note that UBMoE-LLM improves the generation performance on TruthfulQA. Is it possible to provide an additional explanation for the improvements?
>
> We're glad you're paying attention to that. We believe this is mainly due to the introduction of bidirectional experts, which enhances the model’s resistance to hallucination.
>
> > Summary
>
> Thank you for your valuable advice. We have addressed these suggestions through additional experiments and analyses. We hope these revisions strengthen the validity of our findings.

---

### Official Review · Reviewer_u1oZ · 2025-03-13

**Overall Recommendation:** 3

**Summary:**

The paper investigates the impact of bidirectional fine-tuning on unidirectional language models. The authors argue that bidirectional attention mechanisms, while enhancing embedding tasks, degrade generative performance. To address this, they integrate unidirectional FNN layers with bidirectional ones trained via unsupervised contrastive learning. Extensive experiments demonstrate the model's ability to improve embedding performance without compromising generative capabilities.

**Claims And Evidence:**

The primary claim is that bidirectional fine-tuning increases "subsequent dependence," negatively impacting generative performance. Evidence includes ablation studies comparing various fine-tuning strategies and datasets, showing UBMoE-LLM effectively balances embedding and generation tasks. The study also claims the FNN layer is least affected by bidirectional training, supported by experiments revealing minimal impact on generative performance when only the FNN layer is fine-tuned.

**Essential References Not Discussed:**

n/a

**Experimental Designs Or Analyses:**

The experimental results involve models of varying sizes (0.5B to 7B) and controlling factors like learning rate and batch size. However, further experiments with more diverse datasets would enhance the generalizability of the findings.

**Methods And Evaluation Criteria:**

The methodology includes training with bidirectional contrastive learning using LoRA for larger models and testing on diverse datasets like Natural Instructions, DOQA, and Dureader. The evaluation metrics include F1 scores, accuracy, and rouge-l, providing a comprehensive performance assessment.

**Other Comments Or Suggestions:**

- Clarify the distinction between preceding and subsequent dependencies.

- Include a broader comparison with other hybrid models.

**Other Strengths And Weaknesses:**

Strengths:

- Comprehensive evaluation across diverse datasets.

- The introduction of the attention dependence measure is novel and insightful.

Weaknesses:

- Limited discussion of potential trade-offs in training complexity.

- Additional analysis on the interpretability of attention weights could enrich the findings.

**Questions For Authors:**

- How does UBMoE-LLM perform on tasks beyond text generation and embedding, such as reasoning tasks?

- Have you explored the impact of scaling UBMoE-LLM beyond 7B parameters?

**Relation To Broader Scientific Literature:**

The contribution builds on previous works exploring bidirectional modeling in causal language models.

**Theoretical Claims:**

The paper introduces a novel attention dependence measure to quantify preceding and subsequent dependencies. The derivation is consistent with established attention weight calculations, though a deeper examination of convergence guarantees would strengthen the claims.

---

> ### Author Rebuttal · Authors · 2025-04-01
>
> > Weakness 1: Limited discussion of potential trade-offs in training complexity.
>
> Thanks for the helpful comments. Our training consists of two main parts: bidirectional expert training and UBMoE-LLM model training. For bidirectional expert training, compared to previous bidirectional approaches, we only fine-tune the FNN layers, which significantly reduces training costs. As for UBMoE-LLM training, we train for only 1,000 steps, using just 32 samples per step. The training of UBMoE-LLM (7B) can be completed in just 1 hour on a single H100 GPU.
>
> > Weakness 2: Additional analysis on the interpretability of attention weights could enrich the findings.
>
> Thanks for the helpful comments. We analyzed the relationship between generation performance and attention-based forward or subsequent dependence. The experimental results are shown in the table below. As the level of subsequent dependence increased, the generation performance of the model declined. In contrast, our method reduced the subsequent dependence introduced by incorporating bidirectional experts while improving generation performance. This indicates a correlation between the distribution of attention weights and the model's generation performance.
>
> |         Model         |  DPB  |  DPA  | MMLU  |
> | :-------------------: | :---: | :---: | :---: |
> | Qwen2.5-0.5B-Instruct | 55.68 | 39.75 | 46.9  |
> |       UBMoE-LLM       | 54.68 | 38.91 | 43.85 |
> |        FNN-Bi         | 54.40 | 41.40 | 35.54 |
> | Qwen2.5-1.5B-Instruct | 58.27 | 36.57 | 60.21 |
> |       UBMoE-LLM       | 58.08 | 36.80 | 59.41 |
> |        FNN-Bi         | 57.00 | 38.13 | 54.35 |
> |  Qwen2.5-7B-Instruct  | 60.94 | 35.07 | 73.86 |
> |       UBMoE-LLM       | 60.37 | 35.61 | 70.86 |
> |        FNN-Bi         | 46.17 | 49.05 | 21.28 |
>
> > Weakness 3: Clarify the distinction between preceding and subsequent dependencies.
>
> We provide a detailed explanation of the computation methods for preceding and subsequent dependencies in Section 3.2. As shown in formula 1, preceding dependence represents the average score of attention before token i in the attention layer. Subsequent dependence represents the average score of attention after token i.
>
> > Weakness 4: Include a broader comparison with other hybrid models.
>
> Thanks for your advice and we added MNTP-LLM as a baseline, which is trained using the Wikipedia dataset through the MNTP method.
>
> |         Model         | MMLU  | Winogrande | Truthfulqa |  Avg  |
> | :-------------------: | :---: | :--------: | :--------: | :---: |
> | Qwen2.5-0.5B-Instruct | 46.90 |   55.60    |   41.86    | 48.12 |
> |       UBMoE-LLM       | 43.85 |   56.94    |   43.02    | 47.94 |
> |       MNTP-LLM        | 41.33 |   54.40    |   41.65    | 45.79 |
>
> |        Model        | MMLU  | Winogrande | Truthfulqa |  Avg  |
> | :-----------------: | :---: | :--------: | :--------: | :---: |
> | Qwen2.5-7B-Instruct | 73.86 |    73.6    |   64.72    | 70.72 |
> |      UBMoE-LLM      | 70.86 |   75.42    |   62.65    | 69.64 |
> |      MNTP-LLM       | 70.06 |   74.21    |   62.54    | 68.93 |
>
> > Question 1: How does UBMoE-LLM perform on tasks beyond text generation and embedding, such as reasoning tasks?
>
> As shown in the table below, we evaluated the performance of UBMoE-LLM on a physics problem benchmark and a mathematics problem benchmark. The experimental results indicate that UBMoE-LLM still demonstrates performance comparable to that of a causal language model.
>
> | Model                 | Math  | PIQA  | Avg   |
> | --------------------- | ----- | ----- | ----- |
> | Qwen2.5-0.5B-Instruct | 32.5  | 70.62 | 51.56 |
> | UBMoE-LLM             | 32.67 | 69.85 | 51.26 |
> | Qwen2.5-1.5B-Instruct | 41.07 | 76.22 | 58.64 |
> | UBMoE-LLM             | 42.68 | 76.36 | 59.52 |
> | Qwen2.5-7B-Instruct   | 54.51 | 79.49 | 67    |
> | UBMoE-LLM             | 53.91 | 79.04 | 66.48 |
>
>
> > Question 2: Have you explored the impact of scaling UBMoE-LLM beyond 7B parameters?
>
> As shown below, we present the experimental results on Qwen2.5-14B-instruct, where UBMoE-LLM maintains performance consistent with that of a causal language model in terms of generation quality.
>
> | Model                | MMLU  | Winogrande | Truthfulqa | Avg   |
> | -------------------- | ----- | ---------- | ---------- | ----- |
> | Qwen2.5-14B-Instruct | 79.54 | 78.74      | 67.51      | 75.26 |
> | UBMoE-LLM            | 78.44 | 77.83      | 68.01      | 74.76 |
>
> > Summary
>
> Thank you for your advice. Your suggestions have brought great help to the improvement of our work. Our results show that the method efficiently combines both generative and embedding tasks efficiently, which supports our hypothesis of attention dependence. We hope these additional analyses address your concerns and improve the manuscript's rigor. We sincerely appreciate your constructive feedback and have carefully addressed all the points raised. We hope the revised version demonstrates significant improvements and better aligns with your expectations.

---

### Official Review · Reviewer_bS72 · 2025-03-14

**Overall Recommendation:** 2

**Summary:**

Due to the unidirectional attention mechanism, current LLMs underperform in embedding tasks. Some studies have modified the unidirectional attention to bidirectional attention in LLMs and fine-tuned them using contrastive learning, resulting in models better suited for embedding tasks. However, this modification compromises the model's generative capabilities, rendering it ineffective for generation tasks.

The authors propose a two-stage training approach to address this issue. In the **first stage**, a feedforward neural network (FNN_b) suitable for embedding tasks is trained. Specifically, the causal attention is replaced with bidirectional attention, while freezing all model parameters except the FNN, and training embedding tasks using contrastive learning. In the **second stage**, the trained FNN from the first stage is combined with the original FNN, and a gate router is added to form a Mixture of Experts (MoE) model. Then, all parameters except the router are frozen, and auto-regressive training is conducted on generation tasks. This results in a new model that excels in both embedding and generation tasks.

**Claims And Evidence:**

1. The article provides a detailed experimental ablation on the effectiveness of fine-tuning layers of LLM (FFN, embedding, attention) for embedding tasks.

2. The article lacks a discussion on the proposition of using MoE (Mixture of Experts) models for both embedding tasks and generation tasks, such as efficiency or performance, which are not addressed (given the inherent router in both tasks, it is deemed unnecessary in my opinion).

**Essential References Not Discussed:**

None

**Experimental Designs Or Analyses:**

1. What is the experimental setup for downstream task testing in the article? For example, is the model used for the evaluation of embedding tasks taken after the second stage of training or before the second stage of training? Does the embedding task evaluation employ bidirectional attention or causal attention?
2. In the results of Table 3, which layer of the model was used to test **subsequent dependence**, and was unidirectional attention or bidirectional attention used during the evaluation?
3. The experiments seem to lack any comparison with previous state-of-the-art (SOTA) methods.

**Methods And Evaluation Criteria:**

The new model developed using the author's approach has increased computational demands for generation tasks. Moreover, for tasks with clear distinctions such as embedding tasks and generation tasks, it is not particularly necessary to employ a single model to achieve both (unless it is training-free). This is because these two tasks inherently have a natural router (the task category is known in advance), and a more efficient and cost-effective solution can be directly implemented through an if-else branch (eliminating the need for a second training phase, as there is no requirement to train a router).

**Other Comments Or Suggestions:**

None

**Other Strengths And Weaknesses:**

#### Strengths:

- The author's proposal of "attention dependence to explain this phenomenon" in Section 3.1 is quite novel, offering a unique perspective for observing the characteristics of attention.

#### Weaknesses:

- In my opinion, the author's motivation is not sufficiently justified. For details, please refer to **Methods and Evaluation Criteria**.

**Questions For Authors:**

None

**Relation To Broader Scientific Literature:**

None

**Theoretical Claims:**

In Section 3.1, under "Preliminaries," the concepts of "Subsequent Dependence" and "decline of productive ability" are only correlated, but causality cannot be inferred. Therefore, the logic of introducing the MoE architecture based on this point is insufficient.

---

> ### Author Rebuttal · Authors · 2025-04-01
>
> > W1: The new model developed ... computational demands for generation tasks.
>
> UBMoE activates only one expert for each token, adding computational overhead during inference solely for token routing. Compared to the computational cost of the causal language model itself, this additional overhead is minimal. As shown in the table below, our approach does not result in significant additional computational overhead.
>
> | Model                 | GFLOPs   |
> | --------------------- | -------- |
> | Qwen2.5-0.5B-Instruct | 505.82   |
> | UBMoE-LLM (0.5B)      | 505.86   |
> | Qwen2.5-1.5B-Instruct | 1,580.62 |
> | UBMoE-LLM (1.5B)      | 1,580.70 |
> | Qwen2.5-7B-Instruct   | 7,239.97 |
> | UBMoE-LLM (7B)        | 7,240.18 |
>
> > W2:  For tasks with clear distinctions such ... requirement to train a router).
>
> A model with both generation and embedding capabilities can reduce deployment costs by eliminating the need to deploy multiple models.  Meanwhile, the method proposed in this paper can also enhance the model’s resistance to hallucination to some extent by introducing bidirectional experts, which improve the model’s performance against linguistic priors. Some work has reflected the advantages of combining unidirectional  and bidirectional  models \[1\]\[2\]. Our results at Truthfulqa benchmark in Table 4 also show that the bidirectional module effectively reduces the LLM hallucination.
>
> [1] LLM2Vec: Large Language ModelsAreSecretly Powerful Text Encoders
> [2] BatGPT: A Bidirectional Autoregessive Talker from Generative Pre-trained Transformer
>
> > W3:  In Section 3.1, under "Preliminaries," ...  based on this point is insufficient.
>
> As shown in Table 1, there is a consistent trend between subsequent dependence and the general capability of the model. This indicates a correlation between subsequent dependence and the model’s general performance, suggesting a potential causal relationship.
>
> Besides, we analyzed the relationship between generation performance and attention-based forward or subsequent dependence. As shown in the table below. As the level of subsequent dependence increased, the generation performance of the model declined. In contrast, our method reduces the subsequent dependence introduced by incorporating bidirectional experts while improving generation performance. This indicates a correlation between the distribution of attention weights and the model's generation performance.
>
> |         Model         |  DPB  |  DPA  | MMLU  |
> | :-------------------: | :---: | :---: | :---: |
> | Qwen2.5-0.5B-Instruct | 55.68 | 39.75 | 46.9  |
> |       UBMoE-LLM       | 54.68 | 38.91 | 43.85 |
> |        FNN-Bi         | 54.40 | 41.40 | 35.54 |
> | Qwen2.5-1.5B-Instruct | 58.27 | 36.57 | 60.21 |
> |       UBMoE-LLM       | 58.08 | 36.80 | 59.41 |
> |        FNN-Bi         | 57.00 | 38.13 | 54.35 |
> |  Qwen2.5-7B-Instruct  | 60.94 | 35.07 | 73.86 |
> |       UBMoE-LLM       | 60.37 | 35.61 | 70.86 |
> |        FNN-Bi         | 46.17 | 49.05 | 21.28 |
>
> > Q1: What is the experimental setup for ... employ bidirectional attention or causal attention?
>
> We provide detailed experimental settings in Section 5.1. For the evaluation of embedding capability in Table 2, we maintain the same settings as in the training phase and enable bidirectional attention.
>
> > Q2: In the results of Table 3, which layer of ... used during the evaluation?
>
> Following the computation method described in Section 3.2, we use the average across all layers. During the calculation, we adopt causal attention to remain consistent with the generation task.
>
> > W4: The experiments seem to lack any comparison with previous state-of-the-art (SOTA) methods.
>
> Thanks for your advice. We added MNTP-LLM as a baseline, which is trained using the Wikipedia dataset through the MNTP method. As shown in the table below, the experimental results demonstrate that UBMoE-LLM still exhibits strong performance.
>
> |         Model         | MMLU  | Winogrande | Truthfulqa |  Avg  |
> | :-------------------: | :---: | :--------: | :--------: | :---: |
> | Qwen2.5-0.5B-Instruct | 46.90 |   55.60    |   41.86    | 48.12 |
> |       UBMoE-LLM       | 43.85 |   56.94    |   43.02    | 47.94 |
> |       MNTP-LLM        | 41.33 |   54.40    |   41.65    | 45.79 |
>
> |        Model        | MMLU  | Winogrande | Truthfulqa |  Avg  |
> | :-----------------: | :---: | :--------: | :--------: | :---: |
> | Qwen2.5-7B-Instruct | 73.86 |    73.6    |   64.72    | 70.72 |
> |      UBMoE-LLM      | 70.86 |   75.42    |   62.65    | 69.64 |
> |      MNTP-LLM       | 70.06 |   74.21    |   62.54    | 68.93 |
>
> > Summary
>
> Thank you for your helpful advice. We will improve our work based on your suggestions. It is worth noting that the concept of attention dependence proposed by us is highly correlated with the generative ability of the model in experimental verification, and has important significance for the study of unidirectional and bidirectional architecture of the model. We hope our response provides a clearer perspective on our work.

---

### Decision · Program_Chairs · 2025-05-01

**Decision:**

Accept (poster)

**Comment:**

The paper proposes a two-stage training framework to unify the strengths of bidirectional and unidirectional attention for large language models, aiming to improve both embedding and generation capabilities. The authors first train a bidirectional expert for embedding tasks and later integrate it with the original model using a Mixture-of-Experts (MoE) structure, guided by a gating mechanism. One key concern from reviewers was the potential computational overhead introduced by the MoE design, especially during generation. The authors addressed this with detailed GFLOPs comparisons, showing the overhead is negligible. In the response, the authors also responded by highlighting deployment benefits and improvements on hallucination benchmarks, providing expanded empirical evidence to support their attention dependence hypothesis, and added a baseline comparison with MNTP-LLM to address the lack of SOTA comparisons.

The AC has calibrated the reviewers’ comments, particularly for those who did not actively engage during the discussion phase. Overall, the reviewers’ concerns have been adequately addressed, and the authors’ responses are thoughtful and well-supported.